# Aminoglycosides-Related Ototoxicity: Mechanisms, Risk Factors, and Prevention in Pediatric Patients

**DOI:** 10.3390/ph16101353

**Published:** 2023-09-25

**Authors:** Serena Rivetti, Alberto Romano, Stefano Mastrangelo, Giorgio Attinà, Palma Maurizi, Antonio Ruggiero

**Affiliations:** 1Pediatric Oncology Unit, Fondazione Policlinico Universitario Agostino Gemelli IRCCS, 00168 Rome, Italy; serena.rivetti@guest.policlinicogemelli.it (S.R.); alberto.romano@guest.policlinicogemelli.it (A.R.); stefano.mastrangelo@unicatt.it (S.M.); giorgio.attina@policlinicogemelli.it (G.A.); palma.maurizi@unicatt.it (P.M.); 2Dipartimento di Scienze della Vita e Sanità Pubblica, Università Cattolica del Sacro Cuore, 00168 Rome, Italy

**Keywords:** aminoglycosides, children, ototoxicity, antibiotics, antibiotic side effects, childhood

## Abstract

Aminoglycosides are broad-spectrum antibiotics largely used in children, but they have potential toxic side effects, including ototoxicity. Ototoxicity from aminoglycosides is permanent and is a consequence of its action on the inner ear cells via multiple mechanisms. Both uncontrollable risk factors and controllable risk factors are involved in the pathogenesis of aminoglycoside-related ototoxicity and, because of the irreversibility of ototoxicity, an important undertaking for preventing ototoxicity includes antibiotic stewardship to limit the use of aminoglycosides. Aminoglycosides are fundamental in the treatment of numerous infectious conditions at neonatal and pediatric age. In childhood, normal auditory function ensures adequate neurocognitive and social development. Hearing damage from aminoglycosides can therefore strongly affect the normal growth of the child. This review describes the molecular mechanisms of aminoglycoside-related ototoxicity and analyzes the risk factors and the potential otoprotective strategies in pediatric patients.

## 1. Introduction

Aminoglycosides are antibiotics with a broad spectrum of action used in the treatment of numerous infections in all age groups of patients [1].

They are used in the treatment of severe infections of early childhood but are burdened by various side effects, including ototoxicity with hearing loss. In childhood, normal hearing function is of fundamental importance to allow normal cognitive development and language development. Consequently, knowledge of the mechanisms underlying the hearing damage from aminoglycosides, and of the possible preventive strategies, can help to prevent the appearance of this irreversible side effect.

In this review, we analyzed the pathogenetic mechanisms underlying the onset of hearing loss from aminoglycosides, the risk factors, and the protective strategies available in pediatric patients.

### Research Methods

This research aimed to write a narrative review to summarize the knowledge currently available on the pathogenesis and prevention of ototoxicity from aminoglycosides in children. To reach this goal, from 20 January 2023 to 30 June 2023, we searched on Pubmed for papers dedicated to this topic and performed a Pubmed-based retrieval of articles using the search terms “aminoglycosides” and “ototoxicity” matched with “children”. After the original search, we used filters to select articles available in the English language and articles with available full texts. This research retrieved 391 articles. Two operators set the 391 articles according to the adherence of the title and abstract to the topic. At the end of this search, a total of 161 papers were obtained and included in the review.

## 2. Pharmacokinetics and Pharmacodynamics of Aminoglycosides

### 2.1. Aminoglycosides Structure

Aminoglycoside antimicrobials were discovered in the 1940s. The first one, called streptomycin, was isolated from the Gram-positive bacteria *Streptomyces griseus* and used for the treatment of tuberculosis. Since then, other aminoglycosides have been derived from *Streptomyces* species (neomycin, tobramycin, kanamycin, paromomycin, and spectinomycin). Others have been derived from *Micromonospora* species (gentamicin and sisomicin) or from preexisting molecules of aminoglycoside through chemical modifications (netilmicin, amikacin, plazomicin, arbekacin) [2].

Aminoglycoside structures are formed by a hexose (aminocyclitol) bound to one or more aminosugars through glycoside bonds. Streptomycin is made up of a guanidinylated streptamine linked in 4-position to a disaccharide. In the other three classes of aminoglycosides, the aminocyclitol is 2-deoxystreptamine; it can be 4,5-disubstitued (neomycin, neamine, paromomycin, ribostamycin), 4,6-disubstitued (amikacin, arbekacin, kanamycin, tobramycin, gentamicin, sisomicin, netilmicin, plazomicin), or mono-substituted (apramycin, neamine) [2].

Based on the spectrum of action, four generations of aminoglycosides have been identified, according to their ability to evade bacterial resistance mechanisms: generations I (streptomycin, neomycin, kanamycin, monomycin), II (gentamicin), III (tobramycin, amikacin, netilmicin, sisomicin), and IV (isepamicin), and a new generation (plazomicin) [3].

### 2.2. Aminoglycoside Absorption and Distribution

Aminoglycosides are hydrophobic, polycationic molecules, their cationic nature resulting from a predominance of basic, ionizable amino groups in the chemical structure. They have bactericidal activity trough interference with bacterial protein synthesis [2].

Because of their cationic nature, aminoglycosides are not lipophilic and hence have a very poor adsorption via the gastrointestinal tract; consequently, oral administration of aminoglycosides is not recommended. Thus, they need to be administered intramuscularly or, more commonly, by the intravenous route. Aminoglycosides demonstrate remarkably similar kinetics. They exhibit low plasma protein binding (<10%) [1].

After parenteral administration, the volume of distribution of aminoglycosides approaches approximately total body volume, with good distribution in all tissues, mainly into the extracellular water due to their hydrophilic nature. Their apparent volume of distribution (Vd) decreases according to the increasing of age: gentamicin Vd varies from 0.5–0.7 L/kg in preterm infants to 0.25 L/kg in young adults, due to a higher proportion of total body water in neonates [4]. They have good penetration to several bodily fluids, including synovial fluid, peritoneal, ascitic, and pleural fluids, but penetrate poorly into the central nervous system and the vitreous. Aminoglycosides distribute quite slowly into bile, feces, the prostate, and amniotic fluid. Vd increases in conditions such as sepsis, severe burns and febrile neutropenia, congestive cardiac failure, peritonitis, in the immediate postpartum period, and on parenteral nutrition [1].

They enter the cell via electrostatic binding of their positive molecules to the negative components of the bacterial cell surface (lipopolysaccharide and phospholipids of the Gram-negative outer membrane and phospholipids and teichoic acid of Gram-positive bacteria). This binding allows access to the periplasmatic space. Anaerobic bacteria are generally immune to aminoglycosides due to the lack of a membrane potential and the electron transport mechanisms required for the drug uptake. From the periplasmatic space, a small number of aminoglycosides cross the inner membrane and enter the cytoplasm [5]. In the cytoplasm, they impact protein synthesis by inhibiting initiation of translation, blocking elongation of translation or promoting codon misreading; this process leads to the generation of mistranslated proteins, which cause damage to the inner membrane [6]. This event facilitates the uptake of other aminoglycosides, which accumulate within the cell and accelerate the process of mistranslation, resulting in concentration-dependent bacterial killing by aminoglycosides [7].

### 2.3. Aminoglycoside Mechanism of Action

Aminoglycoside bind the 16S rRNA (A-site) of the 30S ribosomal subunit, inducing conformational change of rRNA in the decoding region that results in the misreading of information from mRNA and errors in protein synthesis. They also inhibit ribosome translocation by immobilization of peptidyl-tRNA at the A-site [8].

Some aminoglycosides (kanamycin, neomycin B, and gentamicin) showed bonds to the allosteric site of 23S rRNA of the 50S ribosomal subunit [9]. Unlike others, the binding site of streptomycin seems to be in the immediate vicinity of the ribosomal decoding center, so it interferes with initial tRNA selection [7]. Furthermore, it has been shown to disrupt the formation of the initiating 70S complex, inhibiting the protein-synthesis termination step [10].

The bacterial cell death following aminoglycoside uptake seems to be due to insertion of misread proteins into the inner membrane, which leads to destabilization of the cell; in addition, the massive uptake of aminoglycosides leads to inhibition of ribosomal activity and blocking of the protein synthesis [11,12].

### 2.4. Aminoglycoside Excretion

Aminoglycosides are excreted renally as intact compounds, by glomerular filtration; their clearance is similar to creatinine clearance and is reduced in the setting of poor renal function. Hence, clearance is also reduced in the elderly and in the neonate [13]. Their urinary concentrations are 70% of the dose administered or more; thus, they are ideal for treatment of urinary tract infections [14]. Elimination half-lives are approximately 2–3 h in adults but are prolonged in young children, especially neonates, due to their immature renal function, and in end-stage renal disease [1].

### 2.5. Aminoglycoside Pharmacokinetics in Pediatric Patients

In pediatric patients, pharmacokinetics (PK) and pharmacodynamics of aminoglycosides is different than in adults and changes over the course of age as extracellular fluid representation and renal function change. Neonates and infants have higher extracellular fluids per kilogram than children and adults, and this affect the volume of distribution of water-soluble medications, such as aminoglycosides, resulting in a higher volume of distribution, which decreases with age [15]. Renal elimination is also affected by age; premature neonates have compromised renal function and glomerular filtration rate increases with age and exceeds adult values during childhood, but it gradually decreases to approximate adult values during adolescence [15]. In addition to these two, several other factors may influence the PK of aminoglycosides in pediatric patients, such as obesity, inflammation, organ failure, critical illness, and co-medication; these factors affect drug exposure and consequently conventional age or weight-based dosing regimens do not seem to be optimal. An inadequate dosage can lead to treatment failure or, on the contrary, an exacerbation of the side effects connected to it. In order to prevent the damage from inadequate dosage, as for other drugs, also for aminoglycosides, in recent years, population PK (PopPK) modeling has been developed, which, combined with therapeutic drug monitoring, allows one to adapt the dose to the patient [16]. PopPKs are currently available for pediatric patients for gentamicin, amikacin, and netilmicin, and they highlight that the main variables influencing the PK of aminoglycosides in pediatric patients are the volume of distribution and renal function [17,18,19]. Furthermore, to date, there are models available that analyze the interactions between pharmacokinetic and pharmacodynamic (PK/PD) parameters. In a recent study carried out by Zazo et al., for example, it was observed, through a PK/PD model, that the most frequently used dose of gentamicin may not be adequate in newborns, who may need administration at longer time intervals with a consequent reduction of toxic effects [20]. In addition, Dong et al. exploited a PK model to evaluate the variables that most influence the onset of hearing damage and observed that, in patients with cystic fibrosis undergoing therapy with Tobramycin, the main determinants are represented by repeated cycles and older age (due to the repetition of treatments over time) [21]. To confirm the results of this study, as the authors specified, there is a need for validation in a larger prospective sample. Prospective studies are needed to develop improved physiological-based PK (PBPK) models to predict the concentration of the drug in the inner ear [21]. PBPK modeling is a compartment and flow-based type of pharmacokinetic modeling in which each compartment represents a physiologically discrete entity, such as an organ or tissue, and this is combined with the blood flow into and out of those entities. The distribution of the drug into and out of that organ will be related to the blood flow into and out of that organ, the concentration in the blood, and a partition coefficient. In theory, if a PBPK model contained a compartment for each organ in the body, it could facilitate the simultaneous description of drug concentration changes over time in each organ. The compartments are not limited to entire organs, and often PBPK models contain nested compartments that represent different cell types within an organ, and even different organelles within a cell. These levels of hierarchical complexity permit the modeling of molecularly-driven events, such as specific metabolic pathway damage mechanisms. With a specific inner ear compartment model, the concentration in the inner ear would be more accurately estimated and would provide a more definitive answer as to whether dosage correlates with the amount of hearing loss. To the best of our research, no specific data are currently available on the PBPK model of aminoglycosides and on hearing damage in pediatric age.

## 3. Clinical Indications for the Use of Aminoglycosides

Aminoglycosides are broad-spectrum antibiotics with activity against both Gram-negative and Gram-positive bacteria. Concerning the first ones, they are active against the Enterobacteriaceae family, including *Escherichia coli*, *Klebsiella pneumoniae* and *Klebsiella oxytoca*, *Enterobacter cloacae* and *Enterobacter aerogenes*, *Providencia* spp., *Proteus* spp., *Morganella* spp., *Serratia* spp., *Yersinia pestis*, and *Francisella tularensis* [22]. Concerning the Gram-positive ones, aminoglycosides have good activity against *Staphylococcus aureus*, including methicillin-resistant and vancomycin-intermediate and -resistant isolates, *Pseudomonas aeruginosa* and, to a lesser extent, *Acinetobacter baumannii* [23,24].

Aminoglycoside activity is enhanced through synergy with other classes of antimicrobials, such as beta-lactams, which are active against the cell wall of Gram-negative and Gram-positive bacteria, including wild-type and multidrug-resistant isolates [25]. Therefore, aminoglycosides are often used in combination with beta-lactams for the empirical treatment of severe sepsis or nosocomial infection in patients with a high risk of mortality or when the suspected causative pathogen may be resistant to more commonly used agents [26]. Their use can be considered for patients infected by multidrug-resistant bacteria, such as carbapenem-resistant Enterobacteriaceae. They are also important in the treatment of multidrug-resistant Tuberculosis.

Aminoglycosides are largely use in children with septic shock. They are recommended for empiric broad-spectrum therapy in settings where ceftriaxone resistance is common in Gram-negative bacteria. In children with immune compromise, they can be used in addition to a beta-lactam or a second/third-generation cephalosporin when antibiotic resistance is a concern. This use should also be considered in patients with sepsis at high risk for resistant Gram-negative infections, in addition to a beta-lactam with beta-lactamase inhibitor agent for empiric therapy [27].

Aminoglycosides could be used for the management of children with fever and neutropenia in addiction to an antipseudomonal beta-lactam, a fourth-generation cephalosporin, or a carbapenem for patients who are clinically unstable, or for centers with a high rate of resistant pathogens [28].

Gentamicin use is recommended for the empiric management of suspected neonatal early-onset sepsis, in addiction to ampicillin. Gentamicin dosing is a 4 mg/kg/dose, given intravenously every 24 h [29,30].

This regimen is also adequate for late-onset neonatal sepsis of newborns admitted from the community and therefore at low risk for infection caused by a multidrug-resistant pathogen. In this regimen, gentamicin dosing is 5 mg/kg/dose, given intravenously every 24 h [31].

Aminoglycosides (e.g., gentamicin 7.5 mg/kg/day, given intravenously and divided into three doses) are appropriate first-line parenteral agents for empiric treatment of urinary tract infections in children who require intravenous treatment [32]. Some studies have demonstrated that once-daily parenteral administration of gentamicin (or ceftriaxone) in a day treatment center is safe, effective, and inexpensive [33,34].

## 4. Pathogenesis of Aminoglycosides—Related Ototoxicity

The real incidence of aminoglycosides-related ototoxicity remains unclear due to variable dose regimens and limited and variable diagnostic strategies. Therefore, in several case studies, the incidence varies between 0% and 63% [35,36].

A recent review by Diepstraten et al. shows that hearing loss occurs in up to 57% of children treated with aminoglycosides [37].

Lanvers-Kaminsky et al. described that the severity of ototoxicity is different among aminoglycosides. Neomycin seems highly toxic; gentamicin, kanamycin, and tobramycin seem medium toxic; and amikacin and netilmicin are considered the least toxic aminoglycosides [38].

The toxicity on the inner ear can manifest as cochleotoxicity or vestibulotoxicity. Cochleotoxicity can cause tinnitus and/or sensorineural hearing loss and can lead to deafness. Vestibulotoxicity can occur as vertigo, nausea, nystagmus, and ataxia.

Streptomycin and gentamicin are mainly vestibulotoxic, while amikacin, neomycin, and kanamycin are preferentially cochleotoxic. Tobramicin is equally vestibulotoxic and cochleotoxic [39]. Vestibulotoxicity can occur in up to 60% of treatment courses. Patients often report vestibular issues and tinnitus before the awareness of hearing loss; however, tinnitus is most often subjective, indicating that it cannot be characterized objectively. The incidence of tinnitus is unclear because of the lack of specific diagnostic tests [40].

Table 1 summarizes the aminoglycoside toxicity on cochlea and vestibule.

To exert their cytotoxic effect, aminoglycosides must enter inner ear tissue (Figure 1).

The blood–labyrinth barrier (BLB) protects the cochlear cells from circulating macromolecules and blood cells. It is made up of the endothelial cells of cochlear blood vessels coupled by tight junctions [41]. Within the cochlea, perilymph has an ionic composition typical of other extracellular fluids (i.e., high in Na^+^, low in K^+^, and millimolar Ca^2+^). Aminoglycosides can enter the perilymphatic space from the bloodstream, but they do not easily enter hair cells. Bathing the apical surfaces of hair cells, endolymph has a unique extracellular ionic composition in the mammalian body (i.e., high in K^+^, low in Na^+^, and ~20-μM Ca^2+^). Cochlear, but not vestibular, endolymph has a highly positive potential of ~+80 mV relative to blood or perilymph that is crucial for sensitive hearing. Perilymph and endolymph remain separated by tight junction-coupled epithelial cells [42,43].

Tran Ba Huy et al. studied the kinetics of gentamicin in the fluids and tissues of rats and reported higher aminoglycoside levels in perilymph than in endolymph, suggesting that the accumulation of aminoglycosides in perilymph, from which the half-life of disappearance is extremely slow, was the primary source of toxicity to hair cells [42]. After systemic administration, aminoglycosides enter perilymph, but this does not appear to be the route to the hair cells. In fact, when Li and Steyger tested the systemically administered fluorescently labeled gentamicin, they demonstrated that it was more efficiently taken up by hair cells than labeled gentamicin perfused into the scala tympani. Therefore, when applied systemically, gentamicin seems to readily cross the BLB and clear into the endolymph prior to entering hair cells [43].

The way of entry to the lymph from the capillaries in the stria vascularis is currently not clear, but it may be via ion channels, membrane transporters, or transcytosis [44].

Systemically administered aminoglycosides can enter hair cells crossing the BLB of the stria vascularis in the lateral wall of the cochlea, but if they are infused into the perilymphatic compartment, they overrun the endolymph [45]. Mechanisms by which aminoglycosides cross the endothelial cells of the BLB into the stria vascularis and transverse the tight junction-coupled marginal cells to the endolymph remain unknown. Less is known about how systemically administered aminoglycosides traffic to vestibular hair cells, although the transitional and dark cells seem to take up aminoglycosides more readily than hair cells and their surrounding supporting cells [46]. Aminoglycosides, like other toxins, need to enter hair cells to cause permanent ototoxicity. They entry into hair cells from the endolymph, primarily via receptor-mediated endocytosis, even if other mechanisms can be possible.

The first mechanism of entry is situated on the apical membrane of hair cells; there are mechanoelectrical transduction (MET) channels, which are non-selective cation channels with an asymmetric permeation pathway formed by a wide extracellular-facing vestibule and a narrow selectivity filter. The permeation pathway of the MET channel seems to be formed by a dimer of transmembrane channel-like protein-1 (TMC1) [47,48,49,50].

The MET channels have high permeability but low conductance for calcium (Ca^2+^) ions. The Ca^2+^ ions, with low concentration in the endolymph, bind negative charges in the vestibule of an MET channel and then within the selectivity filter, from where they can move into the hair cell or back out into the endolymph [51]. The entry of Ca^2+^ ions into the hair cells is driven by the potential difference between the endolymph (+80 mV) and the resting membrane potential of cochlear hair cells (from −40 to −70 mV), producing an electrical gradient of 120–150 mV across the apical membrane of the hair cell [47]. The MET channels are large enough to allow aminoglycosides and other large cations to enter the hair cell cytosol. Aminoglycosides seem to compete with the Ca^2+^ ions for binding the channel pore. Once inside hair cells, aminoglycosides appear unable to exit via MET channels, because the intracellular side lacks a vestibule and has a high energy barrier for reentry from the cytosol [47,48,49]. Via depletion of intracellular PIP2, aminoglycosides inhibit voltage-gated potassium channels, preventing repolarization and leading to a sustained cellular depolarization of the hair cell, which likely contributes to hair cell death [52].

Other routes of entry of aminoglycosides into hair cells are less important than MET channels, but they take importance when MET channels are nonfunctional. There is evidence of a role of a family of ion channels called transient receptor potential (TRP) channels in the endocytosis of aminoglycosides [53].

Systemically administered aminoglycosides initially induce the death of the outer hair cells (OHC) of the basal region of the cochlea, used for perception of high frequencies. Continued dosing leads to the death of OHC of the apical regions of the cochlea, used for lower frequencies; in addition, inner hair cells begin to die [54]. The surrounding supporting cells expand between dying hair cells, reforming apical tight junctions between adjacent cells to maintain the structural integrity of the reticular lamina of the organ of Corti, like a “scar” [55].

Once inside inner ear, aminoglycosides cause ototoxicity via multiple mechanisms acting on sensory hair cells or nonsensory cells with homeostatic functions, like on marginal and intermediate cells in the stria vascularis [56,57]. Furthermore, ototoxicity can occur in the neural pathway between the peripheral inner ear and the cortex of the brain, disrupting auditory and vestibular perception.

In the perilymph, aminoglycosides can block the efferent synapses at the base of outer hair cells blocking the cholinergic nicotinic-like receptors (nAchR) by displacing Ca^2+^ from its specific binding sites; this block at the level of the postsynaptic membrane of the outer hair cells disrupts the medial olivocochlear reflex that protects auditory hair cells from exposure to loud sounds [58,59].

Into the cytoplasm, aminoglycosides bind to hundreds of proteins; for most, the consequences are not known. Aminoglycosides also bind to the phosphatidylinositol family of lipids, particularly phosphatidylinositol 4,5-bisphosphate (PIP2), which blocks voltage-gated, outwardly rectifying potassium channels on the basolateral membranes of OHCs. This blockade prevents the rapid repolarization of hair cells crucial for hair cell survival [52,60]. Intracellular aminoglycosides have been implicated in the degradation of presynaptic ribbons in hair cells that may underlie the reported loss of auditory function in cochlear regions, despite the presence of many surviving hair cells [61].

In bacteria, aminoglycosides bind to ribosomal RNA, causing misreading of messenger RNA (mRNA) and consequent accumulation of misfolded proteins, which leads to cellular stress and bacterial lysis [62]. Since aminoglycosides target bacteria, they also readily disrupt mitochondria within cells, causing the release of proapoptotic factors and oxidative enzymes into the cytoplasm and the generation of free radicals [63,64]. In fact, a mechanism of cell toxicity includes endoplasmic reticulum stress and disruption of mitochondrial integrity, causing the generation of toxic reactive oxygen species that lead to cell death, particularly in hair cells [63,65]. Esterberg et al. studied the role of aminoglycosides in zebrafish neuromast hair cells, showing that they dysregulate the endoplasmic reticulum, leading to calcium flux into mitochondria and to the generation of cytotoxic levels of reactive oxygen species [63,66,67].

Selected mitochondrial mutations (predominantly A1555G) in ribosomal RNA result in a higher binding affinity for aminoglycosides and can cause mistranslation of mRNA during protein synthesis, resulting in cell death [68,69,70].

In addition, some aminoglycosides (gentamicin, streptomycin) can act on the composition of the otolithic membrane, changing its ionic composition and causing vestibular toxicity [71,72].

## 5. Genetic Susceptibility to Aminoglycoside Ototoxicity

Aminoglycosides take effect by interacting with the 30S subunit of bacterial ribosome. They bind to the base pairs C1409-G1491 at the A-site of bacterial 16S rRNA and, in this way, they cause protein mistranslation or premature termination of protein synthesis [73].

Ribosomal rRNAs are arranged into two subunits of different sizes. The prokaryotic ribosomal structure is very similar to the eukaryotic one. The first one is a 70S ribosome, formed by a small 30S subunit and a large 50S subunit; the eukaryotic 80S nuclear ribosome is formed by 40S and a 60S subunits [74]. The structural differences between 70S and 80S have been exploited to create antibiotics directed against bacteria, but not harming human cells [75]. Mitochondrial ribosomes are formed by a small 28S and a large 39S subunits and they have a function in the translation of mitochondrial mRNA in mtRNA; the mitochondrial ribosome contains 12S rRNA, the mitochondrial homologue of the prokaryotic 16S, and eukaryotic nuclear 18S ribosomal RNAs. Even if mitochondrial ribosomes are similar to bacterial ones, human cells show low toxicity to a regular dosage of aminoglycosides because of the double membrane, which surrounds mitochondria and blocks the entrance of the drugs into the organelle [75,76,77]. The genetic susceptibility for aminoglycosides-related ototoxicity was investigated. This susceptibility regards the mammalian mitochondrial genome, transmitted exclusively via the female germ line. The most common RNA mutations predisposing to aminoglycosides ototoxicity are showed in Table 2.

Prezant et al. conducted the first genetic study of ototoxic hearing loss, searching for mutations in the rRNA genes of three Chinese pedigrees and a large Arab–Israeli pedigree with maternally inherited non-syndromic hearing loss [78]. They found the 1555A to G mutation in the 12S rRNA gene in patients with aminoglycosides-induced ototoxicity, but not in control subjects. The most well studied mitochondrial DNA mutations related to ototoxic hearing loss are m.1555A to G and m.1494C to T in the 12S rRNA gene of the 39S subunit. The adenine in 1555 position in thw 12S mitochondrial rRNA gene seems to be equivalent to position 1491 in the 16S rRNA gene in *Escherichia coli* [79]. With the mutation 1555A to G, the secondary structure of 12S rRNA changes and becomes closely similar to the corresponding region of 16S in *E. coli*. This mutation determines conformational change in 12S rRNA structure and maybe the new G-C forms a binding site for aminoglycosides at the A-site of rRNA [80]. Several studies have reported the m.1555A to G worldwide: it is most common in the Chinese pedigree, but it was found also in Japanese, Arab Israeli, and USA families, in Mongolia and in Europe [81,82,83,84]. The incidence of this mutation was approximately 33% in two Japanese group, 13% in a Chinese group, and 17% in two groups from the USA and Spain [85,86,87,88,89].

The mutation 1494C to T is less frequent then the 1555A to G. Lu et al. investigated the frequencies of these mutations in Chinese children with hearing loss and discovered a frequency of 3.96% for m.1555A to G and a frequency of 0.18% for m. 1494C to T [82]. The mutation 1494C to T also contributes to hypersensitivity to aminoglycosides; the cytosine in position 1494 in the A-site of 12S rRNA corresponds to A in position 1555; maybe this mutation determines a conformational change that contributes to determining aminoglycoside action [90].

Other mitochondrial rRNA mutations correlated with non-syndromic hearing loss are reported, but these have a low frequency. Zhao et al. showed the mutation 1095T to C in three Chinese families with ototoxic hearing loss [91]. This mutation in the 12S rRNA gene may affect the initiation of mitochondrial protein synthesis [92]. Other mutations in different loci of 12S rRNA remains of uncertain meaning.

The mutation 1555A to G caused a reduction of 28% and 50% in rate of mitochondrial protein synthesis, in lymphocyte cell lines derived from asymptomatic and symptomatic patients, respectively, from an Arab–Israeli family [80,93]. However, no significant difference between the two groups (35% vs. 37%) was found in cybrid cell lines with constant nuclear background, as if nuclear genes also contribute to the biochemical phenotype of cells carrying the mutations.

Some genetic studies have indicated that the variable penetrance of aminoglycoside-related hearing loss may be due to nuclear genetic background. Four nuclear genes have been identified: mitochondrial transcription optimization 1 (MTO1), GTP binding protein 3 (GTPBP3), 5-methylaminomethyl-2-thiouridylate methyltransferase (TRMU), and mitochondrial transcription factor 1 (TFB1M). GTPBP3, MTO1, and TRMU are hypothesized to act by altering the accuracy of interactions between tRNA anti-codon and mRNA codon in the A-site, but it remains unclear how TFB1M modifies aminoglycoside-related hearing loss [94,95,96].

In subjects with the mutations m.1555A to G or m.1494C to T, these are the primary but not the only factors for the development of hearing loss; they make subjects susceptible to acquiring hearing impairment. The mutations occurring in mitochondrial 12S rRNA cause a conformational change of A-site, which compromises the mitochondrial protein synthesis by influencing codon interaction [69]. In addition, these cells show a reduction in ATP generation and an increase in production of reactive oxygen species [97]. Systemic aminoglycosides concentrate in cochlear cells and are taken up into the mitochondria of the hair cells [69]. In the presence of aminoglycosides, in cells carrying one of these mutations, the mitochondrial translation rate may reduce by approximately 30%, below the minimal level required for survival [80,98]. In particular, the mutation 1555A to G increases the binding of aminoglycosides to 12S rRNA and changes the patterns of chemical modifications by dimethyl sulfate in the presence of aminoglycosides. The insufficient protein synthesis, worsened by the accumulation of aminoglycosides, leads to the apoptosis of hair cells. The damage begins at the base of the cochlea and progresses toward the apex, which is from the outer hair cells to the inner hair cells [99].

**Table 2 pharmaceuticals-16-01353-t002:** Most common RNA mutations predisposing to aminoglycosides ototoxicity and their distribution worldwide.

Mutation	Population Studied	References
1555A to G mutation in the 12S rRNA gene	Three Chinese pedigrees and a large Arab–Israeli pedigree with maternally inherited non-syndromic hearing loss	[78]
m.1555A to G and m.1494C to T in the 12S rRNA gene of the 39S subunit	Most common in Chinese pedigree, but also in Japanese, Arab–Israeli, USA families, in Mongolia and in Europe	[81,82,83]
1095T to C mutation in the 12S rRNA gene	Three Chinese families	[91]

## 6. Factors Enhancing the Risk of Aminoglycoside-Induced Ototoxicity

In addition to genetic predisposition, external factors also contribute to enhancing the risk of hearing impairment after exposure to aminoglycosides (Table 3).

A septic condition potentiates the ototoxicity of aminoglycosides: in mice, a sepsis induced by parenteral injection of lipopolysaccharides (LPS) potentiates the degree of hearing loss [44]. The mechanisms by which inflammation exacerbates aminoglycoside-induced cochleotoxicity remain undetermined [100]. TRPV1 is a key channel in the trafficking of aminoglycosides from the blood into the sensory hair cells by mediating their uptake [43]. In vitro, they showed that TRPV1 agonists increased the uptake of gentamicin into hair cells. The activation of Toll-like receptor 4 (TLR4) can potentiate TRPV1 activity, and LPS is a potent TLR4 agonist [101]. Therefore, systemic LPS may activates TLR4 to upregulate cochlear expression of aminoglycoside-permeant TRPV1, facilitating the cellular uptake of aminoglycosides. This may underlie the subsequent exacerbated cochleotoxicity [100] (Figure 2).

In vivo, aminoglycosides administered during an infection lyse bacteria and increase blood levels of immunogens such as LPS, potentiating pro-inflammatory signaling [102].

The inhibition of TRPV1 lead to decreasing levels of pro-apoptotic signal transducer and activation of transcription-1 (STAT1) relative to STAT3, a pro-survival transcription factor that promotes hair cell survival [103,104]. Mutations of TLR4 or TRPV1, which diminish pro-apoptotic STAT1 levels, may underlie the ability of a significant fraction of individuals with cystic fibrosis who maintain typical hearing thresholds despite high cumulative aminoglycoside dosing [105]. Increasing extracellular calcium ions inhibits inward currents mediated by TRPV1, decreased cellular uptake of (fluorescently tagged) gentamicin, and ameliorated drug-induced cytotoxicity [106].

Systemic inflammation also determines vasodilatation of the blood vessels in the stria vascularis, which may enhance cochlear uptake of aminoglycosides [44].

Local infection, such as acute otitis media, can also exacerbate the aminoglycoside ototoxic damage. In animal models, it was demonstrated that ear inflammation up-regulates the uptake of fluorescently tagged gentamicin into hair cells [107].

Another clinical factor is a concomitant renal insufficiency, which decreases the clearance of aminoglycosides and increases their concentration into the blood [108].

A study in children affected by chronic renal failure showed that the occurrence of sensorineural hearing loss was significantly increased in these patients during the administration of furosemide and aminoglycosides [109]. It has been suggested that in chronic renal failure there are electrolyte disturbances that may change the composition of endolymph and may be responsible for the hearing loss [110].

It is clear that patients affected by chronic kidney disease are more prone to develop sensorineural hearing loss [111]. Uremic toxins can cause serial damage in the cochlea [112]. The decrease in the adenosine triphosphatase sodium–potassium pump (Na^+^–K^+^–ATPase) activity, the amplitudes of cochlear potentials, and further reduction in velocity conduction in the auditory nerve lead to hearing impairment. Hemodialysis, although a renal replacement therapy for uremia, is a risk factor for developing sensorineural hearing loss because of the creation of osmotic disequilibrium of endolymph, ischemia, and subsequent reperfusion that may lead to hearing deficiency [110,113,114,115].

The depletion of endogenous antioxidants, typical of a poor nutritional state, may exacerbate the ototoxicity potential of aminoglycosides by enhancing the role of reactive oxygen species (ROS). An animal study showed that the severity of hearing loss depended on pre-existing tissue glutathione levels, which are low in the case of a protein restricted diet. If glutathione levels were restored by dietary supplementation, the hearing loss was significantly attenuated. In contrast, they showed that healthy pigs, without nutritional deficiency, did not benefit from additional glutathione since its defense mechanisms are not suppressed and because a further elevation of glutathione levels is prevented by homeostatic control [116]. In addition, cochlea ischemia and following reperfusion injury may increase aminoglycosides ototoxicity, if they are administered after the transient ischemia [117]. Using a fluorescent gentamicin tracking technique, Lin et al. demonstrated increased uptake of gentamicin after transient cochlear ischemia. Several factors may explain why ischemia contributes to aminoglycoside ototoxicity: it alters the integrity and permeability of the blood–labyrinth barrier via microcirculatory disorders and oxidative stress and it elicits the release of free iron, which may chelate with the aminoglycoside and induce cascades of ROS formation; furthermore, cochlea ischemia enhance apoptotic cellular death through a caspase-dependent pathway [117].

The concomitant administration of some drugs synergically enhances aminoglycosides ototoxicity, e.g., loop diuretics and vancomycin. A study on an animal model demonstrated that a combination of antibiotics (vancomycin and gentamicin) let to a greater risk of ototoxicity [118]. Loop diuretics are other ototoxic drugs [119]. Administration of an aminoglycoside followed by furosemide may increase the risk of ototoxicity; the aminoglycoside interacts with the cell membranes in the inner ear, increasing their permeability. This theoretically allows the loop diuretic to penetrate the cells in higher concentrations, causing more severe damage [120].

In oncological patients, the risk of ototoxicity related to aminoglycoside administration is added to the ototoxic effect of platinum compounds. Hearing damage is one of the main toxic effects of platinum compounds, derived from the degeneration of the hair cells of the ear. This hearing loss is irreversible, bilateral, and sensorineural. Cisplatin is the most ototoxic agent. Its toxicity is greater for higher cumulative doses, for younger ages (the highest risk is for children under five years of age), for bolus infusion, than short infusion [121,122]. The synergic toxic effects of aminoglycosides and cisplatin have been reported in animal studies; combined cisplatin and gentamicin administration in guinea pigs enhanced ototoxic effect when cisplatin was given early in a 14-day gentamicin course [123].

For many years, noise exposures have been known to enhance aminoglycoside cochleotoxicity; subototoxic doses of these drugs become toxic in the presence of loud sound exposure. Noise exposure determines the vasodilatation of blood vessels in stria vascularis, which enhances aminoglycosides uptake into the cochlea. It also increases the uptake of aminoglycosides, upregulating the aminoglycoside-permeant channels expressed by hair cells [124]. Furthermore, noise leads to the formation of ROS and induces outer hair cell death via ROS/5’ adenosine monophosphate activated protein kinase (AMPKα)-dependent pathway [125].

Loud sound exposure up to two months prior to aminoglycosides administration enhances ototoxicity compared with exposure to aminoglycosides alone [126]. If loud sound exposure occurs at least four weeks later, little or no ototoxic synergy is present. Noise exposure within three weeks after drug exposition will increase ototoxic effects, with decreasing severity over time compared with concomitant noise exposure [127,128].

Furthermore, the severity of aminoglycoside ototoxicity varies with the circadian time of daily administration. Aminoglycoside administration during rest time was associated with a greater risk of hearing loss in mice, as if the circadian rhythm influences ototoxicity [129,130].

Finally, a condition associated with an enhanced risk of aminoglycoside-related ototoxicity is cystic fibrosis, an autosomal recessive genetic disorder. Patients are often hospitalized for life-threatening lung infection, requiring intravenous antibiotic therapy, especially with aminoglycosides and glycopeptides for *Pseudomonas aeruginosa*, methicillin-resistant *Staphylococcus aureus*, and other organism. Therefore, people affected by cystic fibrosis undergo many courses of aminoglycosides therapy during their life [131,132]. Cumulative intravenous antibiotic dosing has a significant negative effect on hearing functions in patients with cystic fibrosis [133]. The incidence of acquired hearing loss in patients with cystic fibrosis is unclear. The prevalence in adult patients ranges from 0 to 47% compared to 11–18% in age-matched groups of adults without a history of cystic fibrosis or aminoglycosides exposure [134,135,136].

Patients with cystic fibrosis are chronically colonized by *Pseudomonas aeruginosa*, and thus often require multiple courses of intravenous aminoglycoside antibiotics for the management of pulmonary exacerbations. Because these antibiotics could be given in higher doses less often, several studies were made to assess the effectiveness and safety of once-daily versus multiple-daily doses of intravenous aminoglycosides. The once-daily dose is less nephrotoxic than the multiple daily dosing in children, in term of a lower rise of creatinine over baseline [137,138]. Regarding the ototoxicity, the results of the studies are not so encouraging. A recent Cochrane review revealed that once- and three-times daily aminoglycoside antibiotics are equally effective in the treatment of pulmonary exacerbations of cystic fibrosis. Analyzing the incidence of ototoxicity in the two groups, there was no significant difference in the relative risk between once- and thrice-daily dosing, with a relative risk of 0.56 (95% CI 0.04 to 7.96, moderate-quality evidence) [138].

## 7. Prevention of Aminoglycoside Ototoxicity

Ototoxicity related to aminoglycosides is irreversible because hair cells do not have the ability to regenerate [139]. Therefore, it is critical to identify otoprotective strategies for reducing the risk of hearing loss. Theoretically, the prevention of aminoglycoside ototoxicity may be played at different levels. Several otoprotective agents have been proposed to prevent aminoglycoside-induced hearing loss. However, none of the otoprotectants under development are currently approved for this purpose.

As radical oxygen species play a role in aminoglycoside mechanisms of ototoxicity, blocking the production or the effects of free radicals and proapoptotic factors has been strongly investigated. Molecules inhibiting one of the many steps in the apoptotic cascade are effective in protecting hair cells in vitro and, to some extent, in vivo [140,141]. Aminoglycosides bind to large numbers of intracellular proteins, and it is not yet clear which of these are crucial for cell survival and which ones can sequester the drug to protect cells.

The administration of antioxidants is effective in animal models where aminoglycoside is the ototoxic agent. Both the vestibular and cochlear side effects of various aminoglycosides have been attenuated [142,143]. Aspirin (Salicylate) is considered an antioxidant drug that can sequester free oxygen radicals. The concomitant administration during therapy with aminoglycoside has been seen to reduce the degree and extent of outer hair cell loss [144]. Furthermore, aspirin inhibits the translocation into the nucleus of cytoplasmic NF-kB (via apoptosis in outer hair cells), suggesting an additional mechanism of otoprotection [145,146]. Therefore, aspirin is a cheap antioxidant that may provide a benefit during aminoglycoside therapy.

The antioxidant D-methionine has been demonstrated to play a potential role against gentamicin- and amikacin-induced ototoxicity by a free radical detoxifying mechanism, prevention of hair cell death, and cochlear mitochondrial glutathione level increases [147,148]. A study on guinea pig demonstrated that D-methionine has the ability to decrease tobramycin-induced ototoxicity in a dose-dependent way, without interfering with aminoglycoside antimicrobial action or with serum gentamicin levels [149]. Although D-methionine otoprotective mechanisms have not been fully clarified, it is known to have an antioxidant role and has not demonstrated interaction with 30s ribosomal inhibition.

Protection against aminoglycoside ototoxicity has also been demonstrated by a wide array of antioxidants (e.g., lipoic acid, Coenzyme Q10, vitamin E) [150,151].

Treatment with antioxidants such as Forskolin or N-acetyl cysteine (NAC) reduces noise-induced ROS formation, prevents activation of AMPKα, and thereby attenuates noise-induced losses of outer hair cells and hearing loss, when assessed one hour after completion of noise exposure [125]. The concomitant administration of NAC with aminoglycosides showed safety and otoprotective effects, also in patents with end-stage renal disease [152,153]. However, its routinely use is not accepted because NAC demonstrated antagonistic effects against gentamicin and tobramycin antimicrobial efficacy [154].

Fenofibrate has a promising role in clinical practice to reduce oxidative stress induced by aminoglycosides. It is an activator of peroxisome proliferator-activated receptors (PPAR-α), used to reduce cholesterol levels in patients at risk of cardiovascular disease [155]. In preclinical studies, fenofibrate has demonstrated otoprotective effect against gentamicin-induced hearing loss via the induction of expression of antioxidant enzymes [156].

In guinea pigs, the use of iron chelators reduced both cochlea- and vestibulo-toxicity of aminoglycosides [143].

A recent study tested moringa extract used as an antioxidant for preventing aminoglycoside-induced hearing loss; in vitro evidence showed significant protection from gentamicin-induced hair cell loss via suppression of ROS, preservation of cytochrome oxidase activity, and reduction in caspase production and cell apoptosis [157].

Alternatively, blocking ototoxin entry into the cochlear fluids and hair cells may be another otoprotective strategy. More studies understanding the mechanisms of permeability of the blood–labyrinth barrier are needed. Competitive blockers of the MET channel, the major channel for the entry route of aminoglycosides, are a strategy to consider. A variety of MET channel blockers were observed to protect against hair cell loss in vitro (e.g., amiloride, quinine or curare), but in vivo they are not shown to be therapeutic agents [46]. For example, D-tubocurarine is an MET channel blocker and, in murine outer hair cells in vitro, permeates the MET channel slower than the aminoglycosides; furthermore, it protects hair cells in zebrafish and cochlear cultures [158]. However, this compound has anticholinergic effect in vivo, blocking the middle ear reflex, causing the remodeling of stereocilia and a transient loss of hearing [159].

Furthermore, calcium competes with aminoglycosides as a permeant blocker of MET channels. In mammalian studies, low calcium levels result in the major uptake of aminoglycosides into hair cells and, conversely, higher calcium levels were otoprotective [160,161]. However, in vivo experiments showed contrasting results. Esterberg et al. demonstrated that spikes in calcium levels in the endoplasmic reticulum and in the mitochondria increase mitochondrial respiration and reactive oxygen species production and lead to a collapse in mitochondrial membrane potential, the release of Ca^2+^ from these intracellular stores, and subsequent cell death [66].

A new otoprotective drug may be a cell-penetrating peptide vaccine, GV1001, which was recently shown to alleviate inflammatory responses, oxidative stress, and apoptosis. A recent study in mice showed significantly reduced outer hair cells loss, and consequently hearing loss occurred when the peptide was parenterally co-administered with a dose of aminoglycoside and furosemide [162].

Translating the preclinical mechanisms of otoprotection into clinical practice remains challenging on numerous levels. These strategies, studied in vitro or in vivo, are potentially applicable to clinical practice, but they need to be validated. A primary requirement for the translation of a candidate otoprotectant into clinical practice is that the protective efficacy must not protect bacteria from the bactericidal effects of aminoglycosides.

Another promising approach is to modify ototoxin structure to reduce ototoxicity. For example, apramycin, an aminoglycoside approved for veterinary use, is a wide-spectrum antibiotic with relatively little ototoxicity, because it has very little activity against mitochondrial ribosomes [70]. However, an applicable strategy is to choose the least ototoxic aminoglycosides in the clinical practice. It is known that amikacin is less ototoxic than gentamicin [163]. Similarly, in hair cells, etimicin and amikacin showed less nephrotoxicity and ototoxicity than gentamicin, because of the minimum oxidative potential of this drug [164]. Optimizing an existing aminoglycoside, sisomicin, could be promising, to reduce the number of positive charges that mediate the uptake into hair cells. Therefore, a new drug was synthetized (N1MS) with lower affinity than sisomicin with the MET channels, but it cannot be used in clinical practice because of a narrowed spectrum of antibacterial activity [76].

Surely, an important role for preventing ototoxicity includes antibiotic stewardship to limit the use of aminoglycosides and the risk of bacterial resistance.

Several studies have compared the once-daily administration with the multiple-daily administration of aminoglycosides in term of efficacy and nephrotoxicity. Meta-analyses of these clinical trials have shown a small but statistically improved clinical outcome with once-daily dosing and a significantly lower incidence of nephrotoxicity with once-daily dosing [164,165,166,167,168]. Thus, the once-daily dosing regimens have similar efficacy or a slightly greater efficacy than multiple-daily dosing regimens. Once-daily dosing can also delay the onset of nephrotoxicity compared with multiple-daily dosing if shorter courses of therapy are used (not more than 5 to 6 days).

Evidence has suggested that once-daily administration of aminoglycosides is better tolerated than the conventional schedules (two or three times daily) in adults and children and offers potential pharmacodynamics advantages [169,170].

Because mutations in the 12S rRNA cause a genetic predisposition to develop aminoglycoside-related hearing loss, a familiar medical history of acquired deafness may orient physicians to choose alternative medications in place of aminoglycosides, even if genetic screening tests may avoid the administration in patients carrying the mutations. In the end, the association of aminoglycosides with other ototoxic drugs should be avoided as much as possible.

## 8. Conclusions

Aminoglycosides represent a cornerstone in the treatment of many different infectious diseases, and their use is very frequent at pediatric age. However, they can determine toxic effects, including hearing damage, particularly in children and neonates that are more susceptible to aminoglycosides damage because of slow elimination due to renal immaturity, elevated Vd due to the high water content in the composition of the body, and increased permeability of the blood–labyrinth barrier. These elements make children more susceptible to ototoxic damage than adults, with a consequent greater risk of abnormalities of language development and cognitive development.

The knowledge available to date has highlighted that there are not many strategies capable of preventing aminoglycoside ototoxicity, with the exception of the method of administration and non-association with diuretic drugs. The search for agents that can reduce hearing damage derived from aminoglycosides represents a challenge that is still open, and its victory would allow us to achieve significant improvements in the quality of life in children subjected to aminoglycosides therapy.

## Figures and Tables

**Figure 1 pharmaceuticals-16-01353-f001:**
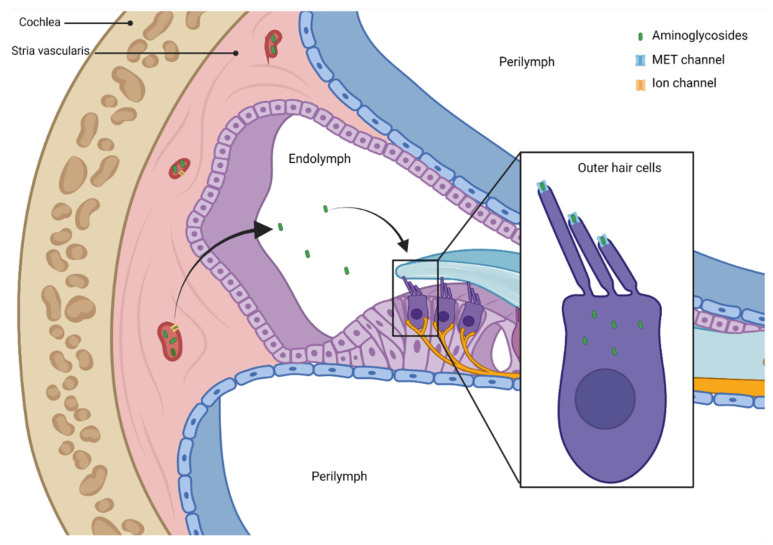
Aminoglycoside damage on ear. Aminoglycosides enter the endolymph from the bloodstream, crossing the endothelial cells of the blood–labyrinth barrier (BLB) into the stria vascularis and the tight junction-coupled marginal cells. They enter into hair cells from the endolymph, primarily via receptor mediated endocytosis. The main mechanism of entry is situated on the apical membrane of hair cells. The mechanoelectrical transduction (MET) channels are non-selective cation channels; aminoglycosides seem to compete with the Ca^2+^ ions for binding the channel pore. Once inside hair cells, aminoglycosides initially induce the death of the outer hair cells (OHC) of the basal region of the cochlea, used for the perception of high frequencies.

**Figure 2 pharmaceuticals-16-01353-f002:**
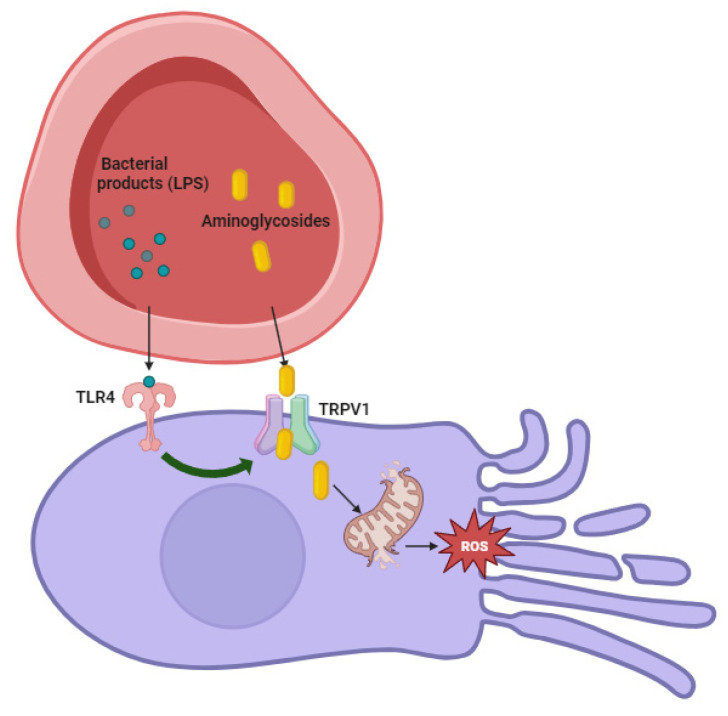
Aminoglycoside damage on ear enhanced by infection. The activation of Toll-like receptor 4 (TLR4) from bacterial products (LPS) can potentiate TRPV1 activity with consequent facilitation of the cellular uptake of aminoglycosides. This may underlie the subsequent exacerbated cochleotoxicity.

**Table 1 pharmaceuticals-16-01353-t001:** Aminoglycoside toxicity on cochlea versus vestibule.

Aminoglycoside	Cochleotoxicity ^a^	Vestibulotoxicity ^b^
Amikacin	yes	Not toxic
Gentamicin	minor	yes
Kanamycin	yes	minor
Netilmicin	yes	yes
Neomycin	very toxic	minor
Streptomicin	minor	very toxic
Tobramycin	yes	yes

^a^ Symptoms of cochleotoxicity: tinnitus, sensorineural hearing loss, deafness. ^b^ Symptoms of vestibulotoxicity: vertigo, nausea, nystagmus, ataxia.

**Table 3 pharmaceuticals-16-01353-t003:** Aminoglycoside ototoxicity-enhancing factors.

Uncontrollable	Controllable
Genetic susceptibility *	Sepsis
Cystic fibrosis (multiple exposures) Neonatal age (high Vd, renal immaturity with low glomerular filtration rate and slow elimination)	Otitis media
	Renal failure (acute or chronic kidney disease)
	Depletion of endogenous antioxidant (e.g., glutathione)
	Cochlea ischemia
	Concomitant ototoxic drugs (e.g., loop diuretics, vancomycin, platinum compounds)
	Loud sound exposure
	Administration during rest time

* 12S RNA gene mutations (m.1555A > G, m.1494C > T, m.1095T > C, others) and nuclear gene mutations (GTPBP3, MTO1, TRMU).

## Data Availability

Data sharing is not applicable.

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
