# Peer review of "Aminoglycosides-Related Ototoxicity: Mechanisms, Risk Factors, and Prevention in Pediatric Patients"

_pharmaceuticals, 2023, doi:10.3390/ph16101353_

Round 1

Reviewer 1 Report

This manuscript offers a comprehensive review of aminoglycoside-induced ototoxicity in the pediatric population, based on an extensive literature analysis.  It discusses various ototoxic mechanisms, interactions with other ototoxic insults, and preventive strategies.

The authors assert that aminoglycoside ototoxicity is more prevalent in the pediatric population due to their frequent use of these drugs.  Moreover, other factors contribute to increased vulnerability among children, including neonates, such as prolonged drug clearance and an underdeveloped blood labyrinth barrier.  In conclusion, understanding these factors is crucial to comprehending the adverse positioning of pediatric patients regarding aminoglycoside-induced ototoxicity.  Please add a paragraph towards the end of the article to summarize these adverse factors.

On page 6, line 262, “Intracellular aminoglycosides have been implicated in the degradation of presynaptic ribbons in hair cells that may underlie the reported loss of auditory function in cochlear regions despite the presence of many surviving hair cells [54].”  The significance of the implication towards ribbon synapses is noteworthy in this context, as aminoglycosides have traditionally been associated with damaging sensory hair cells rather than centrally located neural elements. However, if ribbon synapses are also identified as major targets of aminoglycoside ototoxicity, it would necessitate a revision of the prevailing view and highlight aminoglycosides as potential culprits of "hidden hearing loss." Despite active research in this area, no evidence of HHL-associated with aminoglycosides has been found to date, as indicated by studies with PMIDs: 33667563 and 35806348.

There are quite a few typographical errors throughout the manuscript.  Please proofread more carefully.  Examples:

Line 248: “aging” does not make sense here.  The authors probably want to say “acting”.

Line 373: “toll” rather than “tool”.

Line 436: should be “greater”, and so on.

There are quite a few typographical errors throughout the manuscript.  Please proofread more carefully.  Examples:

Line 248: “aging” does not make sense here.  The authors probably want to say “acting”.

Line 373: “toll” rather than “tool”.

Line 436: should be “greater”, and so on.

Author Response

Reply to Reviewers 1 (in green in the manuscript):

This manuscript offers a comprehensive review of aminoglycoside-induced ototoxicity in the pediatric population, based on an extensive literature analysis.  It discusses various ototoxic mechanisms, interactions with other ototoxic insults, and preventive strategies.

Thanks for the comments. We are glad that this review can be useful in understanding aminoglycoside-induced ototoxicity in pediatric population.

The authors assert that aminoglycoside ototoxicity is more prevalent in the pediatric population due to their frequent use of these drugs.  Moreover, other factors contribute to increased vulnerability among children, including neonates, such as prolonged drug clearance and an underdeveloped blood labyrinth barrier.  In conclusion, understanding these factors is crucial to comprehending the adverse positioning of pediatric patients regarding aminoglycoside-induced ototoxicity.  Please add a paragraph towards the end of the article to summarize these adverse factors.

We added a paragraph in the conclusion to summarize the factors that make children more susceptible to harm from aminoglycosides (lines 641-647).

On page 6, line 262, “Intracellular aminoglycosides have been implicated in the degradation of presynaptic ribbons in hair cells that may underlie the reported loss of auditory function in cochlear regions despite the presence of many surviving hair cells [54].”  The significance of the implication towards ribbon synapses is noteworthy in this context, as aminoglycosides have traditionally been associated with damaging sensory hair cells rather than centrally located neural elements. However, if ribbon synapses are also identified as major targets of aminoglycoside ototoxicity, it would necessitate a revision of the prevailing view and highlight aminoglycosides as potential culprits of "hidden hearing loss." Despite active research in this area, no evidence of HHL-associated with aminoglycosides has been found to date, as indicated by studies with PMIDs: 33667563 and 35806348.

Thank for these relevant comments.

There are quite a few typographical errors throughout the manuscript.  Please proofread more carefully.  Examples:

Line 248: “aging” does not make sense here.  The authors probably want to say “acting”.

Line 373: “toll” rather than “tool”.

Line 436: should be “greater”, and so on.

We corrected them and we checked the manuscript for other typographical errors.

Reviewer 2 Report

Generally speaking, the article is not attractive, and the logic is not clear enough. The article must undergo major revisions before being accepted for publication.

1. Why only search for Pubmed database?

2. Microbial names such as Streptomyces griseus need to be italicized. Please check similar problems through the whole manuscript.

3. line 66-70: This paragraph's logical relationship to the preceding and next paragraphs is not particularly strong. Please check similar problems through the whole manuscript.

4. line 162 and line 174: What is the sample size and dose?

5. The dosage needs to be listed in table 1.

6. The paragraph division of the article is very casual, with some paragraphs only having one sentence.

7. To break down the core point and improve the article's logic, extra subheadings are required.

8. The conclusion must be simplified. The conclusion only needs to express the core idea of the full text.

Moderate editing of English language required

Author Response

Reply to Reviewer 2 (in pink in the manuscript):

Generally speaking, the article is not attractive, and the logic is not clear enough. The article must undergo major revisions before being accepted for publication.

We are sorry that you find the article unattractive. Hearing impairment in pediatric patients is a significant problem because it can cause abnormalities in cognitive development. Given the frequent use of aminoglycosides in children, we think that the knowledge of the mechanisms of the ototoxic damage caused by them may be useful in preventing it.

  1. Why only search for Pubmed database?

Not being a systematic review but a narrative review, we restricted the field to pubmed research only in order to be able to more easily trace the scientific articles that were most significant in achieving our goal.

  1. Microbial names such as Streptomyces griseus need to be italicized. Please check similar problems through the whole manuscript.

We corrected them and checked the manuscript for some other errors like it.

  1. line 66-70: This paragraph's logical relationship to the preceding and next paragraphs is not particularly strong. Please check similar problems through the whole manuscript.

We modified the paragraphs to emphasize the relationship to the preceding and next paragraphs (now it is: “Aminoglycosides are hydrophobic, polycationic molecules. The cationic nature resulting from a predominance of basic, ionizable amino groups in the chemical structure. They have bactericidal activity trough the interference with bacterial protein synthesis [2].

Because their cationic nature, aminoglycosides are not lipophilic and hence have a very poor adsorption via the gastrointestinal tract; consequently, oral administration of aminoglycosides is not recommended. Thus, they need to be administered intramuscularly or, more commonly, by the intravenous route. Aminoglycosides demonstrate remarkably similar kinetics. They exhibit low plasma protein binding (<10%) [1].

After parenteral administration, the volume of distribution of aminoglycosides approaches approximately total body volume, with good distribution in all tissues, mainly into the extracellular water due to their hydrophilic nature.”)

  1. line 162 and line 174: What is the sample size and dose?

Both works are reviews and not original articles. They report the percentages found in the scientific papers they analyzed.

  1. The dosage needs to be listed in table 1.

Table 1 reports the ability of individual aminoglycosides used in therapeutic dosages to cause damage to the cochlea or the vestibule or both structures. We do not think it is appropriate to specify the dosage in this section because it could mislead the reader and induce him to think that the damage is linked exclusively to the dosage while, as highlighted in the manuscript, there are numerous factors that influence the damage from aminoglycosides. In this table we only want to exemplify the structure in which the damage prevails.

  1. The paragraph division of the article is very casual, with some paragraphs only having one sentence.

We revised the article and tried to rearrange some parts.

  1. To break down the core point and improve the article's logic, extra subheadings are required.

We revised the article and added subtitles (particularly in section 2).

  1. The conclusion must be simplified. The conclusion only needs to express the core idea of the full text.

We have modified the conclusions section, also based on the comment of another reviewer. We have tried to underline the characteristics that make ototoxic damage particularly relevant in pediatric patients.

Reviewer 3 Report

This is a review of aminoglycoside ototoxicity that included a structured strategic search with keywords. The authors should clarify if the ototoxicity has been documented in human or animal model (usually rodents), and also compare difference between mouse, rats and guinea pig, since there are interspecies variability.

As a second recommendation, the authors should discriminate between acute and chronic exposure, this is particularly relevant for animal models.

The vestibulo-toxic effect of gentamicin on the otoconia and otolithic membrane in mouse model is not mentioned and these changes are not related to hair cell damaged.

Questions and suggestions:

1. Authors should include the period of study in the search.

2. Can the authors explain the selective cochlea or vestibulo toxicity of each aminoglycoside.

3. Can the authors explain the selectivity for high frequency hearing loss? This deserve a specific comment

4. Regarding mutations in mitochondrial DNA, a Table including mutations, population studies (since mutations changes according to ethnic background) and cochlear and vestibular toxic effects is recommended.

Suggested additional references:

Campos A, López-Escámez JA, Crespo PV, Cañizares FJ, Baeyens JM. Gentamicin ototoxicity in otoconia: quantitative electron probe X-ray microanalysis. Acta Otolaryngol. 1994 Jan;114(1):18-23. doi: 10.3109/00016489409126011. PMID: 8128848.

López-Escámez JA, Cañizares FJ, Crespo PV, Baeyens JM, Campos A. Electron probe microanalysis of gentamicin-induced changes on ionic composition of the vestibular gelatinous membrane. Hear Res. 1994 Jun 1;76(1-2):60-6. doi: 10.1016/0378-5955(94)90087-6. PMID: 7928715.

Author Response

Reply to Reviewer 3 (in blue in the manuscript):

This is a review of aminoglycoside ototoxicity that included a structured strategic search with keywords. The authors should clarify if the ototoxicity has been documented in human or animal model (usually rodents), and also compare difference between mouse, rats and guinea pig, since there are interspecies variability.

As a second recommendation, the authors should discriminate between acute and chronic exposure, this is particularly relevant for animal models.

The vestibulo-toxic effect of gentamicin on the otoconia and otolithic membrane in mouse model is not mentioned and these changes are not related to hair cell damaged.

Thanks for the comments. We tried to answer to your questions, and we added the two additional references to our manuscript.

Below are our responses.

Questions and suggestions:

  1. Authors should include the period of study in the search.

We added it.

  1. Can the authors explain the selective cochlea or vestibulo toxicity of each aminoglycoside.

In table 1 reports the ability of individual aminoglycosides used in therapeutic dosages to cause damage to the cochlea or the vestibule or both structures. Also, in line 201-202 we specified it.

  1. Can the authors explain the selectivity for high frequency hearing loss? This deserve a specific comment

In lines 269-270 we specified that “aminoglycosides initially induce death of the outer hair cells (OHC) of the basal region of the cochlea, used for perception of high frequencies.” Also, in figure 1 we emphasize this concept.

  1. Regarding mutations in mitochondrial DNA, a Table including mutations, population studies (since mutations changes according to ethnic background) and cochlear and vestibular toxic effects is recommended.

We added a table (table 3) to summarize the most relevant RNA mutation implicated in the genesis of aminoglycosides ototoxicity predisposition.

Suggested additional references:

Campos A, López-Escámez JA, Crespo PV, Cañizares FJ, Baeyens JM. Gentamicin ototoxicity in otoconia: quantitative electron probe X-ray microanalysis. Acta Otolaryngol. 1994 Jan;114(1):18-23. doi: 10.3109/00016489409126011. PMID: 8128848.

López-Escámez JA, Cañizares FJ, Crespo PV, Baeyens JM, Campos A. Electron probe microanalysis of gentamicin-induced changes on ionic composition of the vestibular gelatinous membrane. Hear Res. 1994 Jun 1;76(1-2):60-6. doi: 10.1016/0378-5955(94)90087-6. PMID: 7928715.

Round 2

Reviewer 2 Report

Accept in present form

Minor editing of English language required

Author Response

Thanks. 

We have revised the English language